# Dielectric Polarization and Electrorheological Response of Poly(ethylaniline)-Coated Reduced Graphene Oxide Nanoflakes with Different Reduction Degrees

**DOI:** 10.3390/polym12112528

**Published:** 2020-10-29

**Authors:** Yudong Wang, Min Yang, Honggang Chen, Xiaopeng Zhao, Jianbo Yin

**Affiliations:** Smart Materials Laboratory, Department of Applied Physics, Northwestern Polytechnical University, Xi’an 710129, China; 1139055802@mail.nwpu.edu.cn (Y.W.); miny@nwpu.edu.cn (M.Y.); 2017100698@mail.nwpu.edu.cn (H.C.); xpzhao@nwpu.edu.cn (X.Z.)

**Keywords:** poly(ethylaniline)/reduced graphene oxide, dielectric polarization, electrorheology

## Abstract

We prepared poly(ethylaniline)-coated graphene oxide nanoflakes and then treated them with different concentrations of hydrazine solution to form dielectric composite nanoflakes having different reduction degrees of reduced graphene oxide core and insulating polyethylaniline shell (PEANI/rGO). The morphology of PEANI/rGO was observed by scanning electron microscopy, while the chemical structure was confirmed by Fourier transform infrared spectroscopy and X-ray photoelectron spectrometer. The influence of reduction degrees on the conductivity, dielectric polarization and electrorheological effect of PEANI/rGO in suspensions was investigated by dielectric spectroscopy and rheological test under electric fields. It shows that the PEANI/rGO has two interfacial polarization processes respectively due to rGO core and PEANI shell. As the number of hydrazine increases, the conductivity and polarization rate of rGO core increase. As a result, the difference between the polarization rate of rGO core and that of the PEANI shell gradually becomes large. This increased difference does not significantly decrease the yield stress but causes the flow instability of PEANI/GO suspensions under the simultaneous action of electric and shear fields.

## 1. Introduction

Electrorheological fluids (ERFs) are a type of electrically responsive suspensions composed of dielectric particles in non-conducting carrier liquid [1,2]. Without electric fields, ERFs exhibit a low viscous liquid state because the particles are randomly suspended. When an electric field is applied, the particles are polarized due to the difference between the dielectric constant of the carrier liquid and that of particles and, consequently, the particles aggregate into chains between two electrodes. These gap-spanning chains can distinctly increase the apparent viscosity of ERFs by several orders of magnitude and even make ERFs a gel-like solid under the effects of an electric fields of the order of kV/mm. This viscosity change or liquid–solid transformation is rapid and reversible. So, ERFs attract significant attention in many technical applications, such as dampers, microfluidics, robotics, transmissions, and soft actuators [3,4,5,6].

To obtain available ERFs, a key aspect is to develop ER particles with a strong electro-response. For such a purpose, graphene has attracted a great deal of interest as a suitable substrate for developing high-performance ER material due to its anisotropic morphology and excellent electrical property [7,8]. However, neat graphene cannot be directly used in ERFs because its high conductivity is easy to result in dielectric breakdown. In contrary, neat graphene oxide (GO) can be directly used in ERFs. For example, Choi’s group, for the first time, used a direct dispersing method to prepare GO/silicone oil ERFs [9]. Hong and Jang used a solvent transfer method to prepare GO/silicone oil ERFs [10]. Compared to the former, the latter shows higher dispersion stability due to the decreased stacking aggregation of GO. However, the volume faction of these GO ERFs is still low because of the poor compatibility of hydrophilic GO in hydrophobic oil. In addition, due to the negatively charged nature of the surface of GO, the electrophoresis is so strong that the ER effect of GO ERFs is weak. To overcome these problems, we used silica and polyhedral oligomeric silsesquioxane (POSS) to coat or modify the surface of GO and thus increase the dispersion stability and decrease the electrophoresis effect of GO in silicone oil [11,12]. Some other GO composites were also developed [13,14]. However, the ER effect of the GO composites was still relatively weak due to the low polarizability and slow polarization rate of insulating GO. In order to improve ER effect, we used highly conducting reduced graphene oxide (rGO) as core to develop a kind of flake-like rGO/semiconducting polymer composite particles for ERFs [15,16]. It was found that the electric field-induced yield stress of the ERFs based on the flake-like rGO composite particles was about twice as high as that of the ERFs based on pure semiconducting polymer particles. After that, some other flake-like composites, such as graphene/amorphous carbon nanoflakes, were also developed and were found to show enhanced ER effect [17]. Despite the promising prospect as a high-performance ER material, the research of rGO composite ERFs is still not deep. Besides the particle morphology effect, the influence of chemical or physical properties of graphene or rGO core on electric polarization and ER effect of composites is incompletely understood. Unveiling this will merit the material design.

For that purpose, we herein prepared poly(ethylaniline)-coated graphene oxide nanoflakes and then reduced them by different concentrations of hydrazine solution to form dielectric composite nanoflakes having different conductivities of rGO core and insulating polyethylaniline shell (PEANI/rGO). The morphology of PEANI/rGO was observed by scanning electron microscopy, while the chemical structure change was confirmed by Fourier transform infrared spectroscopy and X-ray photoelectron spectrometer. The influence of chemical structure and conductivity on dielectric polarization and ER effect of PEANI/rGO nanoflakes in suspensions was investigated by dielectric spectroscopy and rheological test under electric fields.

## 2. Materials and Methods

### 2.1. Preparation of PEANI/GO Composite Nanoflakes

First, a modified Hummers method was used to prepare GO colloid dispersion in water with a concentration of 0.128 g/10 g [18]. Then, 16.88 g GO colloid dispersion was dispersed in 180 mL 1M HClO_4_ solution by ultra-sonication to form colloid dispersion. After that, 6.30 g 2-ethylaniline was added into the colloid dispersion. After further stirring for 40 min to make 2-ethylaniline fully adsorb onto the surface of GO, 60 mL 1M HClO_4_ solution containing 9.00 g ammonium persulfate (APS) was slowly added into the 2-ethylaniline/GO dispersion under stirring at 0 °C. Finally, the dispersion was reacted for 24 h at 0 °C to form a precipitate. The precipitate was filtered, washed with deionized water and ethanol, and dried at 60 °C to obtain PEANI/GO.

### 2.2. Preparation of PEANI/rGO with Different Reduction Degrees

First, 0.75 g PEANI/GO was dispersed in 375 mL water and the dispersion was heated to 95 °C. Then, different concentrations of hydrazine solution (The amounts of hydrazine in 100 mL aqueous solution are 70 μL, 140 μL, 420 μL, 1400 μL, respectively) were added into the PEANI/GO dispersion. Finally, the dispersion was further heated for 3 min at 95 °C and then was rapidly filtered and washed with deionized water and ethanol, dried at 60 °C for 10 h to obtain PEANI/rGO with different reduction degrees.

### 2.3. Preparation of ERFs

First, the density of PEANI/rGO was measured by the pycnometer method. Here, all samples are about 1.54 g/mL in density. Then, the PEANI/rGO samples were further dried at 100 °C for 48 h in vacuum. Finally, the dried PEANI/rGO were mixed with silicone oil (viscosity 50 cSt, density 0.96 g/cm^3^, dielectric constant 2.7 at 25 °C) to get ERFs with a volume fraction of 4.5 vol %.

### 2.4. Characterization and Measurements

The morphology of samples was observed by scanning electron microscope (SEM, Hitachi TM-3000). The functional groups of samples were analyzed by a Fourier transform infrared spectrometer (FT-IR, JASCO FT/IR-470Plus). The surface chemistry of samples was detected by an X-ray photoelectron spectrometer (XPS, K-Alpha). The thermal properties of samples were analyzed by a thermogravity analyzer (TGA, Netzsch STA449F3). The ER effect of ERFs was measured by a stress-controlled rheometer (Thermal Haake RS600) under DC power supply (WYZ-010 HVDC). The dielectric spectra of ERFs were measured by an LCR analyzer (HP 4284A).

## 3. Results and Discussion

Figure 1A shows SEM images of as-synthesized PEANI/GO sample. Different from dry pure GO that is stacked aggregation structure (see Appendix A), the as-synthesized PEANI/GO composite is not stacked aggregation but the free-standing flake-like shape with a thickness of about 200 nm and a lateral size of 3–10 μm (Figure 1A). This is because before coating, the stacked GO has been disintegrated to form single or few-layer GO nanosheets by ultrasonic treatment in water [19], and then the single or few-layer GO nanosheets are coated by PEANI to form PEANI/GO composites. Figure 1B–D show SEM images of PEANI/rGO samples obtained by treating PEANI/GO with different concentrations of hydrazine. It is seen that the PEANI/rGO samples have the same flake shape and size as PEANI/GO, indicating that the hydrazine reduction does not destroy the flake structure and shape. This may be attributed to the fact that in PEANI/GO composites, the GO core is no longer a stacked structure but is a single or few-layer GO that will not be further disintegrated during reduction. In addition, it is also seen that the PEANI coating is uniform and no free PEANI appears. We have noted that the uniformity of the coating is influenced by several factors, such as the adsorption time of 2-ethylaniline monomer on GO before polymerization, the amount of monomer, and the reaction temperature. If the adsorption time is too short, free granular PEANI will appear. If the adsorption time is too long, the oligomer of 2-ethylaniline will increase. The amount of 2-ethylaniline monomer needs to be suitable. If the amount of monomer is too large, free granular PEANI will appear. In order to slow down the polymerization rate and avoid the formation of free granular PEANI, a relatively low reaction temperature is needed. Here 0 °C is employed. We will report a special work about the preparation of composites composed of GO core coated by semiconducting polymers in different acid circumstances elsewhere.

Figure 2A shows the FT-IR spectra of as-synthesized PEANI/GO and PEANI/rGO obtained by reduction with different concentrations of hydrazine. Because the thickness of the PEANI shell is far larger than that of the GO core, only IR bands of the PEANI shell appear in the spectra. Compared with those of as-synthesized PEANI/GO, the IR bands of PEANI/rGO show significant changes. For example, the wide-band in 4000–2000 cm^−1^ caused by the electron motion in conducting PEANI backbone disappears, the band of C=C stretching vibration in the quinone structure shows a blue shift from 1586 cm^−1^ to 1602 cm^−1^, and the absorption band intensity of the quinone structure at around 1157 cm^−1^ decreases significantly. These changes indicate that the PEANI shell has changed from a conducting emeraldine salt into a non-conducting emeraldine base form [20,21,22,23]. Because it is very difficult to completely remove the PEANI shell by solvent washing (even DMF), however, we cannot directly detect the structure change of GO core. In order to know whether the GO core can be transformed into rGO after hydrazine treatment, we characterized the FT-IR spectra of pure GO after hydrazine reduction as shown in Figure 2B. It is seen that after treating with hydrazine, the oxygen-containing groups (C–OH at 1399 cm^−1^ and C–O at 1053 cm^−1^) on GO have been greatly reduced, the C=C absorption band shows a redshift from 1622 cm^−1^ to 1597 cm^−1^, and the IR band of C–O–C at 861 cm^−1^ disappears. These changes indicate that GO can be reduced into rGO form [24,25,26,27]. In particular, as the amount of hydrazine increases, the reduction degree of rGO increases. Although the PEANI shell may partly protect GO from the reduction with hydrazine in the PEANI/GO composite sample, the TGA trace and dielectric spectroscopy results can support GO core can also be reduced into rGO according to the significant change of TGA curves and the dielectric relaxation peak shift towards high-frequency with the increase of the amount of hydrazine. We will discuss it in the TGA and dielectric analysis sections.

Figure 3 shows the XPS spectra of as-synthesized PEANI/GO and PEANI/rGO obtained by reduction with different concentrations of hydrazine. Because the thickness of the PEANI shell is about 100 nm, XPS can only detect the elemental electron binding energy of the PEANI shell as seen in the full spectra (Appendix A) and high-resolution spectra of C1s, Cl1s, and N1s (Figure 3A–C). It is seen that the as-synthesized PEANI/GO mainly contains C, N, O, and Cl elements. O and Cl come from the doped perchloric acid. N has two states including protonated imine (–N^+^=) nitrogen at 402.5 eV and amino (–NH–) nitrogen at 399.5 eV as displayed in Figure 3C [28]. After hydrazine treatment, the Cl element and the protonated imine (–N^+^=) nitrogen at 402.5 eV disappear completely as shown in Figure 3B,C, indicating that the doping agent has been removed from the obtained PEANI/rGO samples. This also supports that the PEANI shell has changed from conducting emeraldine salt into non-conducting emeraldine base form. Similarly, we characterized the XPS spectra of pure GO after hydrazine reduction as shown in Figure 3D. It is seen that the oxygen-containing groups (COOH, C–O, C–OH) on the surface of GO gradually disappear with the amount of hydrazine, indicating that the GO can be reduced into rGO form and the reduction degree of rGO increases as the hydrazine concentration increases. This is consistent with the FT-IR result.

Figure 4A shows TGA curves of as-synthesized PEANI/GO and PEANI/rGO obtained by reduction with different amounts of hydrazine. It is seen that as-synthesized PEANI/GO possesses two decomposition processes including decomposition of oxygen-containing groups on the surface of GO core at 200–350 °C and decomposition of PEANI shell at 400–550 °C [27,29]. Compared with those of pure GO in Figure 4B and pure PEANI in Figure 4A, however, the temperatures of these two decompositions are higher. This may be because there is strong electrostatic interaction between PEANI and GO. After hydrazine reduction, the obtained PEANI/rGO samples mainly show the thermal decomposition process due to PEANI shell at 400–550 °C and the decomposition process corresponding to oxygen-containing groups of GO core at 200–350 °C becomes more and more unobvious as the amount of hydrazine increases. At the same time, the decomposition temperature of PEANI shell in the PEANI/rGO samples also gradually decreases as the amount of hydrazine increases. These indicate that the oxygen-containing groups are gradually removed or the reduction degree of rGO core gradually increases, which is also decreasing the interaction between PEANI shell and rGO core. The decomposition process of pure GO after treating with different amounts of hydrazine also supports the increase of reduction degree of rGO. As displayed in Figure 4B, the residual weight of rGO gradually increases as the amount of hydrazine increases, indicating the removal of the oxygen-containing groups from the surface of GO or the increase of reduction degree of rGO. In addition, the change of TGA curves of samples before and after hydrazine reduction also supports that the GO core can be reduced into rGO though the PEANI shell may partly protect it. This may be because the semiconducting polymers are easy to be permeated by small hydrazine molecules. The effective reduction of GO core has also been reported in another semiconducting polymer/GO composites [30].

The conductivity of pure GO is (8.40 ± 0.60) × 10^−4^ S/m. As the amount of hydrazine increases, the conductivity of the obtained rGO gradually increases (see Appendix A). The conductivity of as-synthesized PEANI/GO is high about 147 ± 10 S/m. This is because the PEANI shell is conducting HClO_4_-doped emeraldine in salt form. The as-synthesized PEANI/GO cannot be used as the dispersed dielectric particles of ERFs because of the electric short circuit under electric fields. After reduction with hydrazine, however, the conductivities of the obtained PEANI/rGO samples become very low. The conductivities are about (1.09 ± 0.15) × 10^−7^ S/m, (9.02 ± 0.57) × 10^−8^ S/m, (5.11 ± 0.50) × 10^−8^ S/m, and (2.56 ± 0.40) × 10^−8^ S/m for PEANI/rGO obtained by reduction with hydrazine of 70 μL, 140 μL, 420 μL, and 1400 μL, respectively. This is because the PEANI shell has been dedoped to form an insulating emeraldine base, which can provide a very good insulating effect for the conducting rGO core. Thus, the PEANI/rGO can be used as the dispersed dielectric particles of ERFs. We investigate their dielectric polarization characteristic below.

Figure 5 shows the permittivity spectra of ERFs of PEANI/rGO at 25 °C. It is seen that there are two dielectric relaxation processes and they become clearer with the increase of the amount of hydrazine as shown by two loss peaks in the imaginary part of permittivity in Figure 5B. Through comparing the experimental values of relaxation times as shown in Figure 5B with the calculated values by Maxwell–Wagner interfacial polarization formula (see Appendix A), it can be clarified that these two dielectric relaxation processes are attributed to the interfacial polarization of rGO core and the interfacial polarization of PEANI shell. The interfacial polarization is often observed in heterogeneous ERFs, which is important for the ER effect [31]. Further considering the values of conductivities, it can be further clarified that the fast relaxation at high-frequency is due to the interfacial polarization of rGO core and the slow relaxation at low frequency is due to the interfacial polarization of PEANI shell in a carrier liquid (see marked note in Figure 5B). As the amount of hydrazine changes, however, the intensity and rate of these two interfacial polarizations change significantly. To make a quantitative comparison, we used the following dielectric equation (Equation (1)) containing two Cole–Cole terms and a DC conductance term to fit the data in Figure 5 [32,33,34,35].
(1)$$\varepsilon^{*}\left(\omega \right)=\varepsilon^{\prime }+i\varepsilon^{\prime \prime }={\varepsilon^{\prime }}_{\infty }+\frac{\Delta {\varepsilon_{1}}^{\prime }}{1+{\left(i\omega \lambda_{1}\right)}^{\beta_{1}}}+\frac{\Delta {\varepsilon_{2}}^{\prime }}{1+{\left(i\omega \lambda_{2}\right)}^{\beta_{2}}}+i\frac{\sigma }{\varepsilon_{0}\omega }$$
where *ε′*_0_ and *ε′*_∞_ represent the relative permittivity corresponding to the lower and upper limits of the angular frequency in the dielectric relaxation distribution, △*ε′= ε′*_0_ − *ε′*_∞_; *ω* represents the angular frequency, *σ* represents the conductivity, λ = 1/*ω_max_* (*ω_max_* represents the low angular frequency corresponding to the loss peak) represents the relaxation time that reveals the rate of polarization, *β* represents the dispersion index of the relaxation time [33], the subscripts 1 and 2 represent the interfacial polarization processes of PEANI shell and rGO core, respectively. We get the solutions of real part *ε′* and imaginary part ε″ as follows,
(2)$$\begin{array}{c} \varepsilon^{\prime }={\varepsilon_{\infty }}^{\prime }+\Delta {\varepsilon_{1}}^{\prime }\left(\frac{1+\left(\omega \lambda_{1}{)}^{\beta_{1}}\cos(\frac{\pi \beta_{1}}{2}\right)}{1+2\left(\omega \lambda_{1}{)}^{\beta_{1}}\cos(\frac{\pi \beta_{1}}{2}\right)+{\left(\omega \lambda_{1}\right)}^{2\beta_{1}}}\right)\\ \;+\Delta {\varepsilon_{2}}^{\prime }\left(\frac{1+\left(\omega \lambda_{2}{)}^{\beta_{2}}\cos(\frac{\pi \beta_{2}}{2}\right)}{1+2\left(\omega \lambda_{2}{)}^{\beta_{2}}\cos(\frac{\pi \beta_{2}}{2}\right)+{\left(\omega \lambda_{2}\right)}^{2\beta_{2}}}\right) \end{array}$$
(3)$$\begin{array}{c} \varepsilon^{\prime \prime }=\Delta {\varepsilon_{1}}^{\prime }\left(\frac{{\left(\omega \lambda_{1}\right)}^{\beta_{1}}\sin(\frac{\pi \beta_{1}}{2})}{1+2\left(\omega \lambda_{1}{)}^{\beta_{1}}\cos(\frac{\pi \beta_{1}}{2}\right)+{\left(\omega \lambda_{1}\right)}^{2\beta_{1}}}\right)\\ \;+\Delta {\varepsilon_{2}}^{\prime }\left(\frac{\left(\omega \lambda_{2}{)}^{\beta_{2}}\sin(\frac{\pi \beta_{2}}{2}\right)}{1+2\left(\omega \lambda_{2}{)}^{\beta_{2}}\cos(\frac{\pi \beta_{2}}{2}\right)+{\left(\omega \lambda_{2}\right)}^{2\beta_{2}}}\right)+\frac{\sigma }{\varepsilon_{0}\omega } \end{array}$$

Figure 5 shows that the fitting curves can well coincide with the permittivity data. Table 1 lists the dielectric characteristic parameters obtained by the fitting. It is seen that the polarization rates of rGO core and PEANI shell are close for PEANI/rGO obtained by reduction with 70 μL of hydrazine. As the amount of hydrazine increases, the polarization rate of the rGO core gradually separates from that of the PEANI shell. With the increase of amount of hydrazine, the interfacial polarization rate of the PEANI shell slows down and the polarization intensity decreases. This should be caused by the decrease of the delocalization ability of charge carriers or the decrease of conductivity of PEANI shell after dedoping. On the contrary, with the increase of amount of hydrazine, the polarization rate of high-frequency interfacial polarization becomes fast and the polarization intensity increases. This should be because of the increase of the delocalization ability of charge carriers or the increase of conductivity of rGO core due to the gradual removal of surface oxygen-containing groups. It also supports that the GO core has been reduced into conducting rGO according to the significant relaxation peak shift towards high frequency with the increase of the amount of hydrazine because the interfacial polarization rate depends on conductivity [31]. In addition, by plotting the reciprocal of λ (λ^−1^) as a function of 1000/*T* (see Appendix A), we can get the activation energy (*E*_a_) of charge movement contributing to the interfacial polarization with Arrhenius equation $\lambda^{-1}\propto \;\exp\left(-\frac{E_{a}}{RT}\right)$, where *R* is the gas constant and *T* is the absolute temperature. From Table 1, it is seen that the value of *E*_a1_ of charge movement contributing to the interfacial polarization of PEANI shell increases and becomes saturated with the increase of amount of hydrazine, also supporting the delocalization ability of charge carriers decreases due to hydrazine dedoping. However, the value of *E*_a2_ of charge movement contributing to the interfacial polarization of rGO core gradually decreases with the increase of amount of hydrazine, also supporting the delocalization ability of charge carriers gradually increases due to hydrazine reduction.

Based on the above characterization and dielectric analysis, we have obtained dielectric composite nanoflakes having different conductivities of rGO core and insulating PEANI shell through treating PEANI/GO with different concentrations of hydrazine solution. The PEANI/rGO ERFs have two interfacial polarization processes respectively due to rGO core and PEANI shell. As the amount of hydrazine increases, the conductivity and the polarization rate of the rGO core increase. As a result, the difference between the polarization rate of rGO core and that of the PEANI shell gradually increases. We further investigate the influence of property change of rGO core on the ER effect of ERFs of PEANI/rGO nanoflakes by rheological measurement under electric fields.

Figure 6 shows the flow curves of the shear stress-shear rate of ERFs of PEANI/rGO without and with electric fields. It is seen that the ERFs are low viscous without electric fields and their off-field viscosities are almost identical. This should be due to the fact that these four PEANI/rGO samples come from the same as-synthesized PEANI/GO and the reduction of different amounts of hydrazine does not affect the size and shape. When electric fields are applied, four ERFs show significant ER effect, i.e., the shear stress increases with the increase of electric field strength. There are distinct differences in the rheological properties among these four samples.

Let us firstly analyze the yield stress. Here, the yield stress is static yield stress (*τ*_s_) is an important parameter to characterize the ER effect of ERFs under electric fields before flow [16,36]. Figure 7A plots *τ*_s_ as a function of electric field strength. It is seen that the value of *τ*_s_ increases with the electric field strength, but the difference of *τ*_s_ among these samples is relatively small. As the amount of hydrazine increases, the value of *τ*_s_ slightly decreases.

After yield, the ERFs start to flow. However, it is seen that the stability of flow is significantly different among these four samples. The flow of ERF of PEANI/rGO obtained by reduction with 70 μL hydrazine is relatively stable, and the shear stress only shows a slight decrease with shear rate and then increases. As the amount of hydrazine increases, the decrease of shear stress of ERFs of obtained PEANI/rGO as a function of shear rate becomes more and more significant as displayed in Figure 6B–D. To compare this more clearly, we extract the shear stress difference Δ*τ* with and without electric fields at the shear rate of 60 s^−1^, which almost corresponds to the shear rate of valley value of shear stress. It can be seen from Figure 7B, as the amount of hydrazine increases, Δ*τ* gradually decreases and becomes lower compared to *τ*_s_. This indicates that the flow becomes more unstable as the amount of hydrazine increase.

It is known that the ER effect originates from the polarization of particles and consequent particle chain structure under electric field stimuli [37,38,39]. Before flow or at yield point, the particle chain structure is complete because no hydrodynamic force disturbs. As a result, ERFs approximately behave as an elastic solid and the yield stress is dominated by the interparticle electrostatic interaction induced by electric fields. In this case, the available polarizability or polarization intensity of particles is important [40]. From Table 1, it is seen that with the increase of amount of hydrazine, the polarization intensity of rGO core increases but that of PEANI shell decreases and, as a result, the total polarization intensity of PEANI/rGO slightly decreases. This is in accordance with the decrease in yield stress. Therefore, the reason for the slight decrease of *τ*_s_ as the amount of hydrazine increases should be attributed to the slight decrease of the total polarization intensity of PEANI/rGO as the amount of hydrazine increases.

After yield, ERFs start to flow, and the flow behavior depends on the balanced situation where the particle chains are continuously broken and rebuilt by the competition between interparticle electrostatic interaction and hydrodynamic interaction. In this case, not only the polarization intensity but also the polarization rate of particles are important because the latter is related to the stability of interparticle interaction or the reorganization of broken chain structure under the simultaneous effect of both electric and shearing fields [40]. Under DC electric fields, it is proposed that to achieve a good flow performance, the relaxation time of ERFs should be located within or near an appropriate range of 1.6 × 10^−3^–1.6 × 10^−6^ s [40,41,42,43]. A too-slow or too-fast polarization rate is not favorable to the stability and the reorganization of chain structure during flow. This is because too slow polarization rate easily results in an insufficient particle polarization during flow, while too fast polarization rate easily results in an increase of repulsive interaction between particles due to the difference between the polarization direction and the direction connecting two particles. From Table 1, it is seen that the relaxation times of PEANI shell and rGO core in PEANI/rGO obtained by reduction with 70 μL hydrazine are 1.36 × 10^−4^ s and 3.00 × 10^−5^ s, respectively. They are not only very close but also located in the appropriate range of 1.6 × 10^−3^–1.6 × 10^−6^ s. Thus, under the simultaneous action of both electric and shearing fields, the particles can maintain sufficient polarizability and a stable interparticle interaction. As a result, the ERF has a relatively stable flow behavior as shown in Figure 6A. As the amount of hydrazine increases, however, the relaxation times of PEANI shell and rGO core gradually separate as displayed in Figure 5B and in Table 1. Meanwhile, the relaxation time of the PEANI shell slowly becomes long but it is still in or near the range of 1.6 × 10^−3^–1.6 × 10^−6^ s, while the relaxation time of the rGO core rapidly becomes short and gradually exceeds the appropriate range. Thus, although the suitable polarization rate of the PEANI shell can maintain the stability of interparticle interaction, but the rapid polarization rate of rGO core tends to increase the repulsive interaction between particles with the increase of shear deformation rate. As a result, the chain structure starts to become unstable and the shear stress decreases as a function of the shear rate as shown in Figure 6B–D. Therefore, the reason for the decrease of flow stability with the increase of the amount of hydrazine should be attributed to the fact that the increases of reduction degree and conductivity of rGO core are resulting in the rapid increase of polarization rate of rGO core and the separation between polarization rates of PEANI shell and rGO core.

By combining the dielectric spectra and rheological property, we can conclude that not only polarizabilities of both shell and core but also the matching level of polarization rates of both shell and core are important to achieve a good ER effect and flow stability of rGO composite. In order to well-match the polarization rates or relaxation times of core and shell in the appropriate range, we should control the rGO core to have a suitable conductivity and to make the dielectric composites have a good conductivity uniformity.

## 4. Conclusions

We prepared PEANI/rGO dielectric composite nanoflakes with different conductivities of rGO core and insulating PEANI shell by treating PEANI/GO with different concentrations of hydrazine solution. The reduction treatment does not change the flake shape and size, but the conductivity of PEANI/rGO has been changed significantly. As the amount of hydrazine increases, the conductivity of the rGO core gradually increases, while that of PEANI shell has been decreased. The change of conductivity has a strong influence on the dielectric polarization and ER effect of PEANI/rGO composite nanoflakes. As the amount of hydrazine increases, the interfacial polarization rate of the rGO core rapidly increases and separates from the polarization rate of the PEANI shell. This separation does not significantly change the yield stress but causes the flow instability of ERFs of PEANI/GO. Our result indicates that to control rGO core to have an appropriate conductivity and thus to make the core-shell structure have a good conductivity uniformity is important for good ER effect and flow stability under DC electric fields.

## Figures and Tables

**Figure 1 polymers-12-02528-f001:**
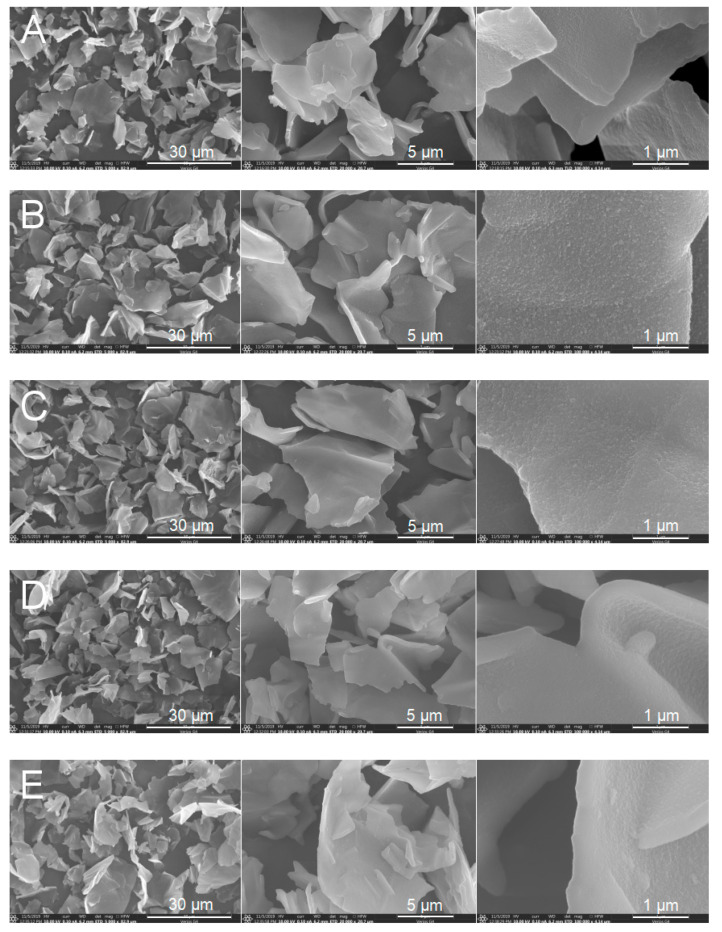
Scanning electron microscope (SEM) images of as-synthesized polyethylaniline shell (PEANI)/graphene oxide (GO) (**A**) PEANI/reduced graphene oxide (rGO) obtained by reduction with hydrazine of 70 μL (**B**), 140 μL (**C**), 420 μL (**D**) and 1400 μL (**E**).

**Figure 2 polymers-12-02528-f002:**
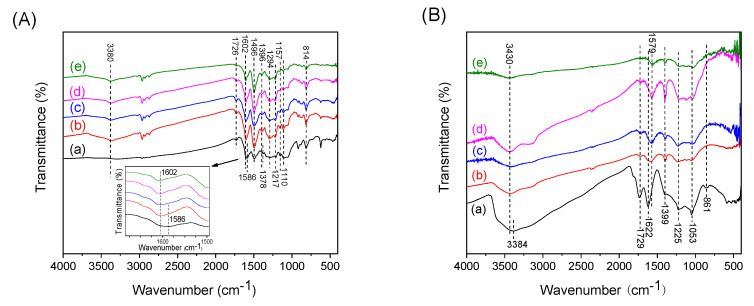
(**A**) FT-IR spectra of as-synthesized PEANI/GO (a) and PEANI/rGO obtained by reduction with hydrazine of 70 μL (b), 140 μL (c), 420 μL (d), and 1400 μL (e). (**B**) FT-IR spectra of GO (a) and rGO obtained after reduction with hydrazine of 25 μL (b), 50 μL (c), 150 μL (d), and 400 μL (e).

**Figure 3 polymers-12-02528-f003:**
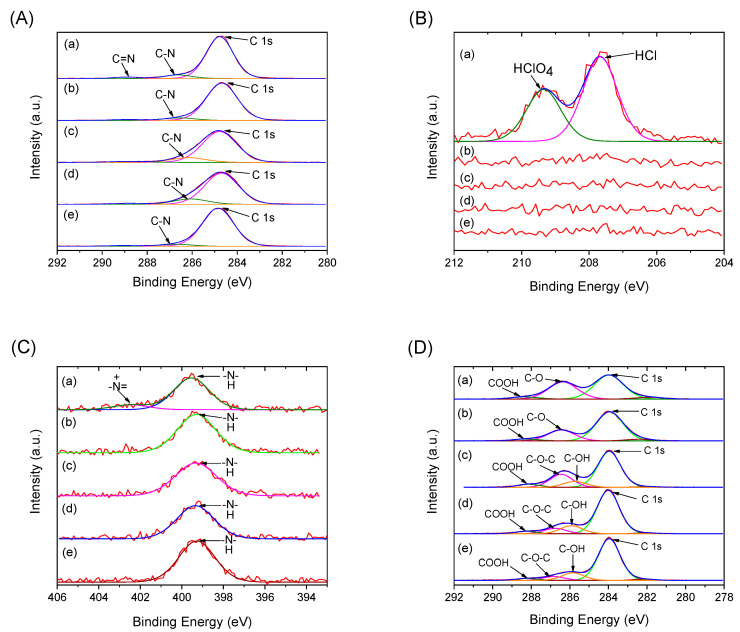
High-resolution X-ray photoelectron spectrometer (XPS) spectra of C 1s (**A**), Cl 1s (**B**) and N 1s (**C**) of as-synthesized PEANI/GO (a) and PEANI/rGO obtained by reduction with hydrazine of 70 μL (b), 140 μL (c), 420 μL (d), and 1400 μL (e); high-resolution XPS spectra of C 1s (**D**) of pure GO (a) and rGO obtained by reduction with hydrazine of 25 μL (b), 50 μL (c), 150 μL (d), and 400 μL (e).

**Figure 4 polymers-12-02528-f004:**
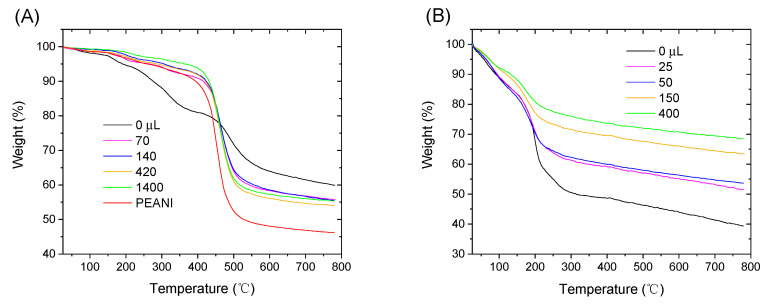
Thermogravity analysis (TGA) curves of (**A**) PEANI/GO and PEANI/rGO obtained by reduction with different amounts of hydrazine; (**B**) pure GO and rGO obtained by reduction with different amounts of hydrazine. TGA curves of pure PEANI are shown in (**A**).

**Figure 5 polymers-12-02528-f005:**
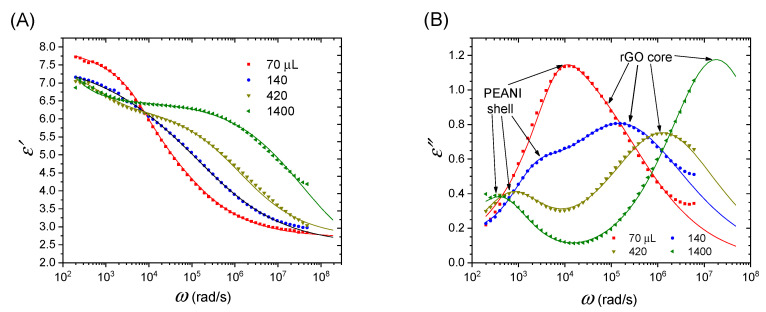
Permittivity spectra of electrorheological fluids (ERFs) of PEANI/rGO obtained by reduction with different amounts of hydrazine at 25 °C: (**A**) real part *ε′*, (**B**) imaginary part *ε*″.

**Figure 6 polymers-12-02528-f006:**
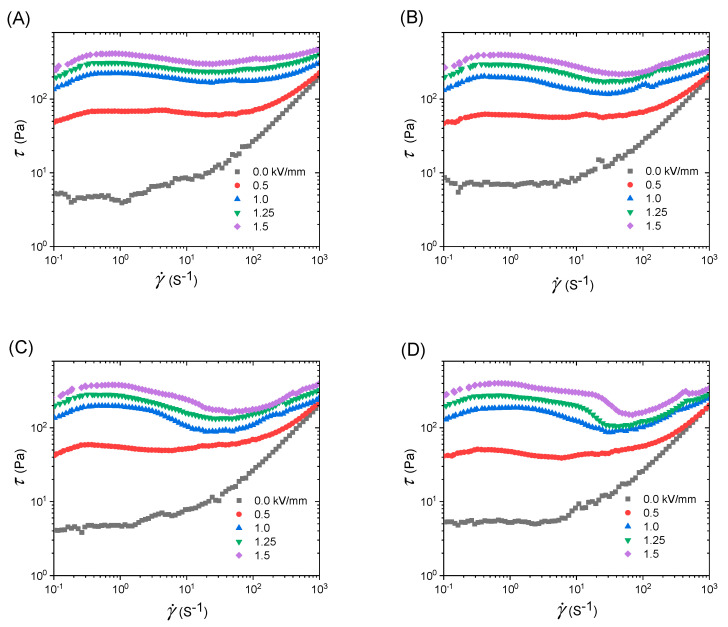
Flow curves of the shear stress-shear rate of ERFs of PEANI/rGO obtained by reduction with different amounts of hydrazine: (**A**) 70 μL, (**B**) 140 μL, (**C**) 420 μL, (**D**) 1400 μL (*T* = 25 °C).

**Figure 7 polymers-12-02528-f007:**
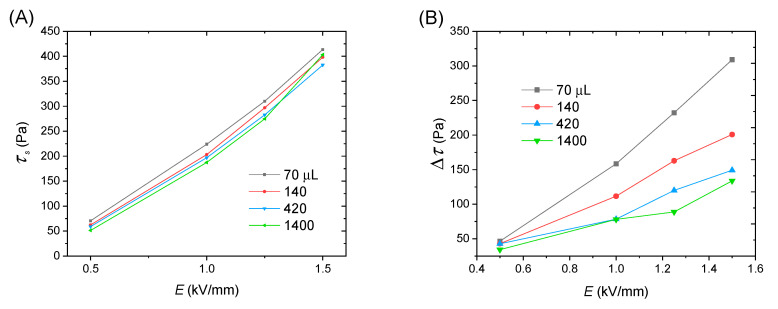
Electric field dependence of static yield stress (**A**) and shear stress difference with and without electric fields at 60 s^−1^ (**B**) for ERFs of PEANI/rGO obtained by reduction with different amounts of hydrazine (*T* = 25 °C).

**Table 1 polymers-12-02528-t001:** Dielectric parameters of ERFs of PEANI/rGO obtained by reduction with different amounts of hydrazine at 25 °C.

Hydrazine (μL)	△*ε*′_1_	△*ε*′_2_	△*ε*′	λ_1_ (S)	λ_2_ (S)	*E_a_*_1_ (kJ/mol)	*E_a_*_2_ (kJ/mol)
70	1.75	3.27	5.02	1.36 × 10^−4^	3.00 × 10^−5^	30.8	33.5
140	1.17	3.49	4.66	4.50 × 10^−4^	6.00 × 10^−6^	52.4	37.0
420	0.89	3.46	4.35	1.50 × 10^−3^	7.80 × 10^−7^	50.7	22.9
1400	0.48	4.50	4.98	2.30 × 10^−3^	5.50 × 10^−8^	47.8	7.1

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
