# Peer review of "Dielectric Polarization and Electrorheological Response of Poly(ethylaniline)-Coated Reduced Graphene Oxide Nanoflakes with Different Reduction Degrees"

_polymers, 2020, doi:10.3390/polym12112528_

Round 1

Reviewer 1 Report

Authors need to check the text again, because the paper has a lot of wrong English sentences. Especially the conclusion part needs to be written again.

Below is a couple of examples :  

     Line 49 : have been use -> have been used

     Line 77 : Predicate – precipitate

     Line 80 : 0.75 g of PEANI/GO -> a PEANI/GO of 0.75 g

The conclusion part (from line 319 ) should be re-written. It has a lot of wrong English sentences.  

Line 114 : you mentioned “Fig. 2A shows FT-IR spectra of PEANI/rGO.”, but the Fig. 2A caption says PEANI/GO. You have to correct it.

Fig (1) through (4) captions have the same problem.:  PEANI/GO. ->  PEANI/rGO.  

Line 118 and 119. You claim that the band of C=C stretching vibration in the quinone structure also has a blue shift from 1586 cm-1 to 1602 cm-1, and the absorption band intensity of the quinone structure at around 1160 cm-1 decreases significantly.

  • It is hard to see the blue shift you claim in the FT-IR data Fig 2 (A). You need to magnify the region in order to make sure the presence of the clear blue shift.
  • The 1160 peak is not observed, you need to put a mark on the position of the peak.

Line 182 -185 : Authors observed two loss peaks in the imaginary part regarding the interfacial polarization process between PEANI shell and rGO core and between PEANI shell and silicone oil.

  • You need to put a mark on the position of the peaks with a short note in the spectra.

Line 230 :

In the Flow curves of shear stress-shear rate in Fig 6,

You have to compare the flow curves to the control ( a PEANI/rGO without hydrazine treatment) and reanalyze the curves with the control curve in order to support your conclusion.

Line 293, and Line 300 and 301, I can not understand what you want to say with the data Fig 6A. Maybe you are talking about Fig 7 A.

Reviewer 2 Report

The paper shows electro/rheological properties of an interesting rGO/PEANI system. However, the manuscript is not yet in the stage that should be submitted. First, it is written with a very poor English with a lot of errors that make reading sometimes quite difficult, and it contains many not enough clear statements. Second, Interpretation of the results is not convincing.

Some examples below:

l. 52 The nature of observed weak electrorheological properties is really a slow polarization rate of insulating GO. Is not it just a smaller polarisibility of GO?

l. 54 The statement “To effectively employ the anisotropic morphology and excellent electrical properties of graphene, we have used reduced graphene oxide (rGO).. “ seems to be rather contradictory. It actually means that for  exploitation of good electrcial properties of some material we have to use something else. The authors should mention here which particular properties of GO are expected to help to achieve the required properties.

l. 103: It is written that “If the adsorption time is too short, impurities will appear.” It is difficult to imagine how the reaction time can influence the concentration of built-in impurities. What kind of impurities do the authors mean?

There should be visible scale bar in Figure 1 and Figure S1. It is difficult to discuss the situation before and after reduction when it is not shown.

If the stacks are assumed to be disintegrated after the reduction with hydrazine the coverage of rGO flakes might not be complete. Please, discuss.

There was no polymer formed in the reaction mixture that was free, not adsorbed on GO?

Hydrazine can reduce both GO and polyaniline.  The reduction of GO has been proved only without the presence of PEANI. It is not clear if the PEANI shell will not protect GO from the reduction with hydrazine. Please, discuss.

l. 155-156: It is written that “Fig. 4A shows the TGA curve of PEANI/rGO with different reduction degrees. It is seen that as-synthesized PEANI/GO possesses two decomposition …“ So, what is really shown in this Figure?

If the dielectric spectrum is to be interpreted as Maxwell Wagner relaxation the constants introduced in the equations 1-2 should be related to conductivity and dielectric constants of the components. Please, discuss.

I think the paper requires major improvements and could be reconsidered again later, after resubmission.

Reviewer 3 Report

Dear author,

The manuscript polymers-946216, entitled ‘Dielectric Polarization and Electrorheological Response of Poly(ethylaniline)-Coated Graphene Oxide Nanoflakes With Different Reduction Degrees’ presents the prepared poly(ethylaniline)-coated graphene oxide nanoflakes and the formation of dielectric composite nanoflakes with different conductivities based on PEANI/rGO. Scanning electron microscopy was used for the morphology study of PEANI/rGO, while the Fourier transform infrared spectroscopy and X-ray photoelectron spectrometry were utilized to confirmed chemical structure changes. In order to study the electrical / dielectrically properties of this materials, in terms of conductivity and dielectric constant, rheological testing under different electric fields and dielectric spectroscopy were used by the authors.

Overall, the manuscript is well written, with good correlation between the obtained results. The manuscript is well written, with good quality images and graphs and supporting tables. The conclusions are in detailed and sustained by the experimental findings. In conclusion, the authors claimed that the key to the achieve good matching of polarization rate or relaxation time of core and shell in suitable range for good ER effect and flow stability is to control rGO core and to make core-shell structure have good conductivity.

In my opinion, this manuscript could be considered for publication in POLYMERS journal AS IS.

Round 2

Reviewer 2 Report

The authors have improved the manuscript markedly. It could be already published but there are still some objections/advises:

  1.  English should be still improved.
  2.  I just wonder what is the accuracy and reproducibility of the conductivity values that are shown with the 3 valid digits accuracy. Please, provide the statistical deviations for these values.
  3.  In the comments to the SEM picture (Fig. 1) it is written that „..the as-synthesized PEANI/GO composite is not stacked aggregation but free-standing nanoflake with a thickness of about 200 nm and a lateral size of 3-10 μm (Fig. 1A). This is because that before coating the stacked GO has been disintegrated to form single or few layer GO nanosheets by ultrasonic treatment .."
    There is actually no proof that the stacked GO has been disintegrated. There are only flakes already covered by the polymer seen in the Figure and it is not clear what is inside. Thickness of 200 nm is enough to digest many GO sheets (thickness of about 1 nm) even if the polymer shell is present.
